# Enabling fast-charging selenium-based aqueous batteries via conversion reaction with copper ions

Chunlong Dai[1], Linyu Hu[1], Hao Chen[2], Xuting Jin[1], Yuyang Han[1], Ying Wang[1], Xiangyang Li[1], Xinqun Zhang[1], Li Song[1], Maowen Xu [2], Huhu Cheng [3], Yang Zhao [1], Zhipan Zhang[1✉], Feng Liu [4✉] & Liangti Qu [1,3✉]

Selenium (Se) is an appealing alternative cathode material for secondary battery systems that recently attracted research interests in the electrochemical energy storage field due to its high theoretical specific capacity and good electronic conductivity. However, despite the relevant capacity contents reported in the literature, Se-based cathodes generally show poor rate capability behavior. To circumvent this issue, we propose a series of selenium@carbon (Se@C) composite positive electrode active materials capable of delivering a four-electron redox reaction when placed in contact with an aqueous copper-ion electrolyte solution (i.e., 0.5 M CuSO$_4$) and copper or zinc foils as negative electrodes. The lab-scale Zn||Se@C cell delivers a discharge voltage of about 1.2 V at 0.5 A g$^{-1}$ and an initial discharge capacity of 1263 mAh g$_{Se}^{-1}$. Interestingly, when a specific charging current of 6 A g$^{-1}$ is applied, the Zn||Se@C cell delivers a stable discharge capacity of around 900 mAh g$_{Se}^{-1}$ independently from the discharge rate. Via physicochemical characterizations and first-principle calculations, we demonstrate that battery performance is strongly associated with the reversible structural changes occurring at the Se-based cathode.

[1] Key Laboratory of Cluster Science, Ministry of Education, Beijing Key Laboratory of Photoelectronic/Electrophotonic Conversion Materials, School of Chemistry and Chemical Engineering, Beijing Institute of Technology, Beijing 100081, P. R. China. [2] Key Laboratory of Luminescent and Real Time Analytical Chemistry (Southwest University), Ministry of Education, School of Materials and Energy, Southwest University, Chongqing 400715, P. R. China. [3] Key Laboratory of Organic Optoelectronics & Molecular Engineering of Ministry of Education, Department of Chemistry, Tsinghua University, 100084 Beijing, P. R. China. [4] State Key Laboratory of Nonlinear Mechanics Institute of Mechanics, Chinese Academy of Sciences, Beijing 100190, P. R. China. ✉email: zhipan@bit.edu.cn; liufeng@imech.ac.cn; lqu@mail.tsinghua.edu.cn

Lithium-sulfur (Li–S) batteries have been considered as one of the most promising energy storage systems as sulfur cathodes show merits of high theoretical specific capacity and low cost[1–3]. However, sulfur suffers from its low intrinsic electronic conductivity, which leads to low active material utilization and unsatisfactory rate performance[4–6]. In addition, the low density of sulfur also prevents it from achieving high areal/volumetric electrochemical performance in real devices[7–9]. Selenium (Se) is a chalcogen element that also shows many advantages in electrochemistry[10–13]. For instance, it has a significantly higher intrinsic conductivity than sulfur ($1 \times 10^{-3}$ vs. $5 \times 10^{-28}$ S m$^{-1}$)[14,15] and a similar theoretical volumetric capacity with sulfur (3253 mAh cm$^{-3}$ for Se, 3467 mAh cm$^{-3}$ for S, all based on a two-electron transfer chemistry)[16,17]. Unfortunately, the low redox potential of Se cathodes makes them only compatible with alkali metal anodes (Li, Na, and K) to construct non-aqueous Li/Na/K–Se batteries[18–29]. These batteries not only get limited by the low mass-specific capacity of Se (theoretical value is 675 mAh g$_{Se}^{-1}$ based on a two-electron transfer chemistry), but suffer from the shuttle effect induced by soluble polyselenides intermediates formed during the charging/discharging process. Such a shuttle effect leads to the continuous loss of active materials and the irreversible consumption of the metal anode, resulting in a fast device degradation and poor cycling performance.

To address these issues, we herein report an aqueous Se cathode chemistry with redox-active Cu$^{2+}$ ions as the charge carriers. The Se cathode executes a four-electron reaction through the sequential conversion of Se $\leftrightarrow$ CuSe $\leftrightarrow$ Cu$_3$Se$_2$ $\leftrightarrow$ Cu$_{2-x}$Se $\leftrightarrow$ Cu$_2$Se, therefore doubling its theoretical gravimetric specific capacity to 1350 mAh g$_{Se}^{-1}$. Even based on the final discharging product (Cu$_2$Se), the theoretical specific capacity is still as high as 517 mAh g$_{Cu2Se}^{-1}$. The step reactions and the total reaction are as following:

$$\text{Step 1}: \text{Se} + \text{Cu}^{2+} + 2e^- \leftrightarrow \text{CuSe} \tag{1}$$

$$\text{Step 2}: \text{CuSe} + 0.5\,\text{Cu}^{2+} + e^- \leftrightarrow 0.5\,\text{Cu}_3\text{Se}_2 \tag{2}$$

$$\text{Step 3}: 0.5\,\text{Cu}_3\text{Se}_2 + (0.5 - x)\text{Cu}^{2+} + 2(0.5 - x)e^- \leftrightarrow \text{Cu}_{2-x}\text{Se}\,(x < 0.5) \tag{3}$$

$$\text{Step 4}: \text{Cu}_{2-x}\text{Se} + x\text{Cu}^{2+} + 2x\,e^- \leftrightarrow \text{Cu}_2\text{Se} \tag{4}$$

$$\text{Total reaction}: \text{Se} + 2\text{Cu}^{2+} + 4e^- \leftrightarrow \text{Cu}_2\text{Se} \tag{5}$$

Meanwhile, the novel Se chemistry shows a high redox potential of about 0.5 V vs. SHE, which is about 1.5 V higher than those of conventional Se cathodes paired with Li/Na/K anodes[18–29]. The enhancement of redox potential could be attributed to the low solubility of CuSe (The solubility product constants (Ksp) of CuSe is $7.9 \times 10^{-49}$)[30]. Based on the Nernst equation, the redox potential of the Se chemistry could be described as follows:

$$E_{Se^0/Se^{2-}} = E^o_{Se^0/Se^{2-}} + \frac{0.059}{2} lg \frac{1}{[Se^{2-}]} \tag{6}$$

$E_{Se^0/Se^{2-}}$ and $E^o_{Se^0/Se^{2-}}$ represent redox potentials of Se$^0$/Se$^{2-}$ in non-standard and standard states; [Se$^{2-}$] represents the effective concentration of Se$^{2-}$ ions. The redox potential could enhance about 1.42 V compared to that in standard state due to the low Se$^{2-}$ ion ($1.6 \times 10^{-48}$ mol/L) in this system. As the intermediates (CuSe, Cu$_3$Se$_2$, and Cu$_{2-x}$Se) are insoluble in the aqueous electrolyte, the shuttle effect of polyselenides observed in alkali metal-Se batteries is circumvented. The Se-based cathode material we propose stores about 800 mAh g$_{Se}^{-1}$ in 5 min. Theoretical simulations show that the high electronic conductivity of the

intermediates/discharging product[31,32] and the accelerated copper ions diffusion/phase transition aroused by the large volumetric deformation during the conversion reaction are responsible for the observed rapid charging/discharging rates. An aqueous Zn||Se@C full cell is then prepared by using a Se@C composite at the cathode, Zn as the anode, Cu$^{2+}$ and Zn$^{2+}$ ions as charge shuttling ions. Interestingly, using a Se@C composite with 48 wt.% of Se, the Zn||Se@C full cell shows an initial discharging capacity of 1263 mAh g$_{Se}^{-1}$ at 0.5 A g$^{-1}$ with a stable discharging plateau at ~1.2 V and can be efficiently charge/discharge for 400 cycles at 2 A g$^{-1}$.

## Results

**Synthesis and characterizations of the Se@C composite.** A honeycomb-like porous carbon was used as the Se host, and the Se@C composite was obtained after a melt-diffusion treatment (Supplementary Fig. 1). As shown in Fig. 1a, the X-ray diffraction (XRD) pattern of Se@C composite is matched with crystalline Se (JCPDS: 06-0362). The Raman spectrum shows two intense peaks at 138.8 and 236.9 cm$^{-1}$, which are assigned to Se (Fig. 1b)[21]. The specific surface area of the Se@C composite is 173 m$^2$ g$^{-1}$, which is lower than that of bare carbon host (370 m$^2$ g$^{-1}$, Fig. 1c). Thermogravimetric analysis (TGA) shows the Se content in Se@C is about 48% in mass (denoted as Se@C-48, where the number represents the Se content, Supplementary Fig. 2), which is comparable to other Se-based batteries reported in the literature[18,20–22,25,26]. Field-emission scanning electron microscopy (FESEM) and transmission electron microscopy (TEM) images show the Se@C-48 composite still maintained the honeycomb-like structure and no large Se particles (diameter is <100 nm) are detected (Fig. 1d–f), suggesting a uniform distribution of Se in the porous carbon host. The element mappings also demonstrate this point (Fig. 1g–i).

**Electrochemical performance of Cu | 0.5 M CuSO$_4$ | Se@C coin cells.** The electrochemical performance of the Se cathode chemistry was studied in the coin cell, with Se@C-48 on carbon cloth as the cathode, a 0.5 M CuSO$_4$ solution as the electrolyte, and a Cu foil as the anode, respectively. The CV curves of Cu|0.5 M CuSO$_4$ | Se@C-48 cells show three pairs of redox peaks (Supplementary Fig. 3). The cells using stainless steel as the cathode current collectors show similar cyclic voltammetry (CV) and galvanostatic charge-discharge (GCD) curves with that using carbon cloth (Supplementary Fig. 4). Considering its flexibility, carbon cloth was adopted as the cathode current collectors in the subsequent electrochemical tests.

Figure 2a shows the GCD curves of the Cu|0.5 M CuSO$_4$ | Se@C-48 cells at 0.5 A g$^{-1}$. It shows a discharging plateau at about 0.16 V vs. Cu$^{2+}$/Cu (~0.50 V vs. SHE), which is more than 1.5 V higher than non-aqueous organic-based alkali metal||Se batteries (Supplementary Fig. 5 and Supplementary Table 1). The first discharge specific capacity is as high as 1298 mAh g$_{Se}^{-1}$, which is close to twice of the conventional Se-based batteries. In addition, the discharge plateau is very stable, releasing about 90% of its full capacity with a charge/discharge voltage hysteresis of about 0.1 V (Fig. 2b). The voltage plateau slope, about 0.11 V/(1000 mAh g$^{-1}$), is smaller than conventional Se cells, means a more stable discharging plateau (Supplementary Fig. 6a). Besides, the hysteresis between discharge and charge curve is also smaller than other Se-based cells (Supplementary Fig. 6b), suggesting reversible reaction kinetics. The Cu|0.5 M CuSO$_4$ | Se@C-48 cell delivers 1070, 1025, 939, 839, 796 mAh g$_{Se}^{-1}$ at 0.5, 2, 5, 8, and 10 A g$^{-1}$, respectively (Figs. 2c, d). The discharge capacity at 10 A g$^{-1}$ is still larger than the theoretical capacity (675 mAh g$^{-1}$) of other Se-based cathodes based on a two-electron transfer reaction. It also shows good cycling stability. A capacity retention

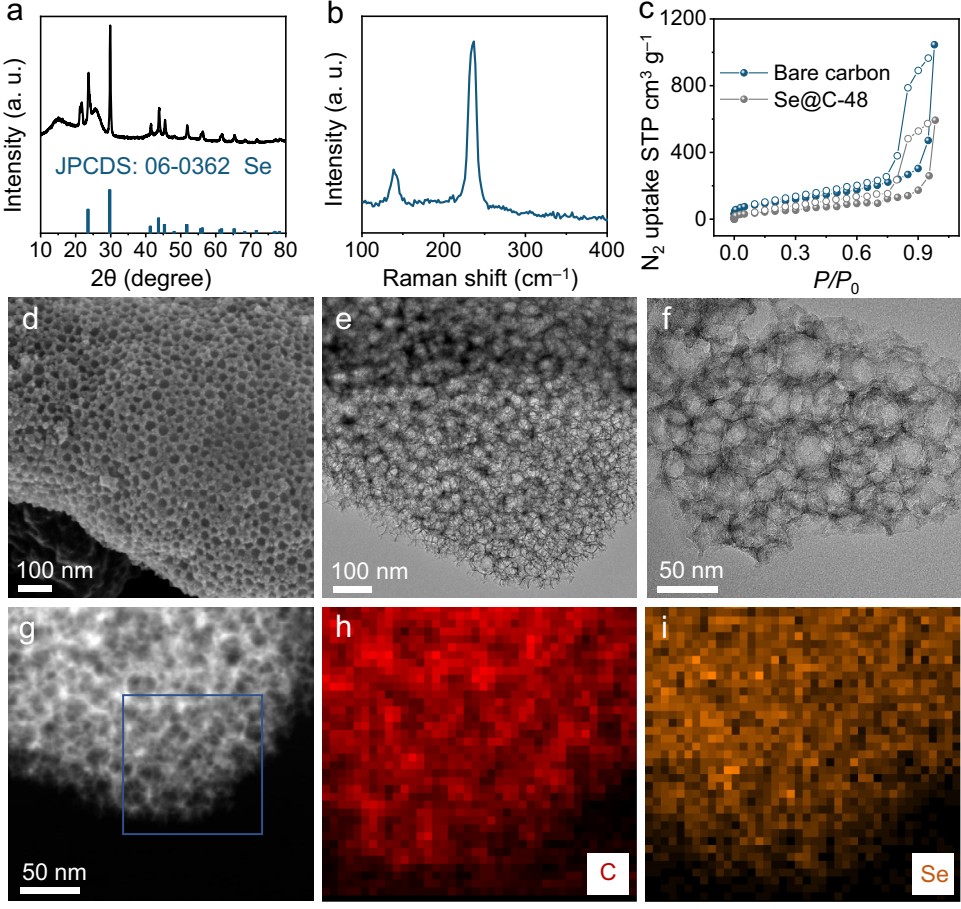

**Fig. 1 Physicochemical characterizations of the Se@C-48 composite. a** XRD pattern, **b** Raman spectrum, and **c** $N_2$ adsorption-desorption isotherm of Se@C-48 composite. The **d** FESEM, **e**, **f** TEM images, and **g–i** corresponding element mappings of the Se@C-48 composite.

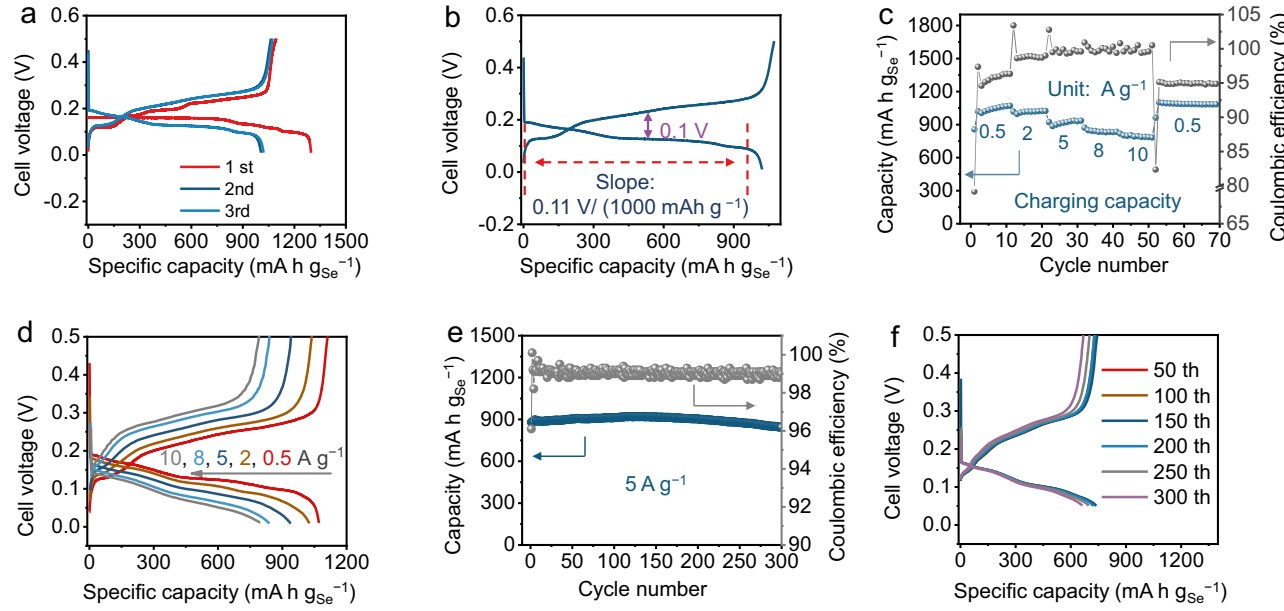

**Fig. 2 Electrochemical characterization of the Cu|0.5 M CuSO$_4$|Se@C-48 coin cells. a** GCD curves of the first three cycles of Cu||Se@C-48 cell (the number represents the Se content) at 0.5 A g$^{-1}$. **b** The discharging plateau and polarization between discharging and charging curves of Cu||Se@C-48 cell at 1 A g$^{-1}$. **c** Rate performance, **d** corresponding GCD curves, **e** cycling stability test (5 A g$^{-1}$), and **f** GCD curves at different cycles of the Cu||Se@C-48 cell.

of about 95% is achieved after 300 cycles at 5 A g$^{-1}$, corresponding to a low decay rate of 0.017% per cycle (Fig. 2e). The GCD curves of different cycles maintain almost the same shape, further demonstrating the good stability (Fig. 2f). The Coulombic efficiency during cycling stability test is >98% except the initial three cycles (~96%). The long-term cycling performance is a result from the fact that no soluble intermediates form during the discharging/charging process, thus hindering the polyselenides shuttle effect present in non-aqueous metal∥Se batteries. After prolonged cycling (300 cycles at 5 A g$^{-1}$), the structure of cathode is well maintained and some Cu$_2$O particles appear on Cu anode due to dissolved oxygen in electrolyte (Supplementary Fig. 7 and 8)[33]. The bare Se demonstrates similar CV and GCD curves as the Se@C-48 cathode, but showing lower specific capacity and rate performance due to large size of bare Se particles (>500 nm, Supplementary Figs. 9 and 10). The pure carbon shows negligible specific capacity (about 1.5 mAh g$^{-1}$) in this system (Supplementary Fig. 11).

Fast-charging batteries that could be charged in a few minutes are highly desired for practical uses. The fast-charging rate performance of this Se chemistry are reported in Fig. 3a. The Cu∥Se@C-48 cell was charged at 10 A g$^{-1}$ and discharged at different specific currents of 0.5, 2, 4, 8, and 10 A g$^{-1}$, respectively. The charging process can be finished in 5 min, and the battery delivers almost the same capacity of about 800 mAh g$_{Se}^{-1}$ when discharging at specific currents of 0.5, 2, 4, 8, and 10 A g$^{-1}$ (Fig. 3b), where the discharging time is 5613, 1450, 732, 365, and 292 s, respectively (Fig. 3c). The effect of Se content in the Se@C composite is further studied (Supplementary Figs. 12–15). Typically, for conventional Se-based batteries, the specific capacity of Se@C composite decreases with the increasing Se content[18–29]. Nevertheless, for this Se chemistry, the charging and discharging capacities remain the same as the Se content increases from 40% to 65%, demonstrating the fast reaction kinetics (Supplementary Figs. 16 and 17). When the Se content increases to 78%, it still delivers discharging capacities of 754, 753, 709, 585, and 546 mAh g$_{Se}^{-1}$ at 0.5, 2, 5, 8, and 10 A g$^{-1}$, respectively (Supplementary Figs. 18 and 19). All Se@C cathodes with different Se contents show small charge transfer resistances

of about 3 Ω (Supplementary Fig. 20 and Supplementary Table 2). Moreover, the fast-charging rate capability performance tests are well-positioned (especially at specific currents >5 A g$^{-1}$) in terms of specific discharge capacity when compared to other Se-based battery chemistry reported in the literature (Supplementary Fig. 21). In order to understand the fast-charging performance, galvanostatic intermittent titration technique (GITT) test is carried out for Cu∥Se@C-78 cell. Typical gaps between each polarization potential and each quasi-equilibrium potential are as small as 15.5 and 18.9 mV for the discharging and charging processes, respectively (Supplementary Fig. 22), suggesting the fast reaction kinetics. The calculated diffusion coefficients of Cu$^{2+}$ are in the ranges of 10$^{-11}$ to 10$^{-8}$ and 10$^{-12}$ to 10$^{-8}$ cm$^2$ s$^{-1}$ for discharging and charging processes, respectively (Fig. 3d,e). The electrochemical performance of Se@C-78 with higher areal loading of about 4, 8, and 12 mg cm$^{-2}$ were also studied (Supplementary Figs. 23–25), where the highest areal capacity of 5.68 mAh cm$^{-2}$ was achieved at an areal loading of 8 mg cm$^{-2}$. Further increasing the areal loading to 12 mg cm$^{-2}$ led to a decrease of the areal capacity to 4.77 mAh cm$^{-2}$ (Supplementary Fig. 26), presumably due to the impeded mass transfer within the thick electrode that detrimentally affect the electrochemical energy storage performance.

**Working mechanism of the Cu∥Se@C coin cells.** In order to study the working mechanism of the Se@C-based electrodes, in situ XRD measurements were performed to examine the Se cathode during a whole discharging-charging cycle. Figure 4a, b show full patterns of in situ XRD between 10 and 55°. There are three peaks that remain constant throughout the process, which come from the XRD equipment (21.6°) and current collect (43.8 and 50.8°)[33]. Before discharging, the XRD pattern of cathode is well matched with crystalline Se (JCPDS: 06-0362) and shows a strongest peak at about 29.7°. The peaks between 25.5 and 30.5° are carefully scrutinized. As shown in Fig. 4c, d, along with the discharging, the peak of Se at 29.7° weakened and disappeared gradually. A peak at about 27.9° associated with the (112) plane of CuSe emerged, suggesting the formation of CuSe by the reduction

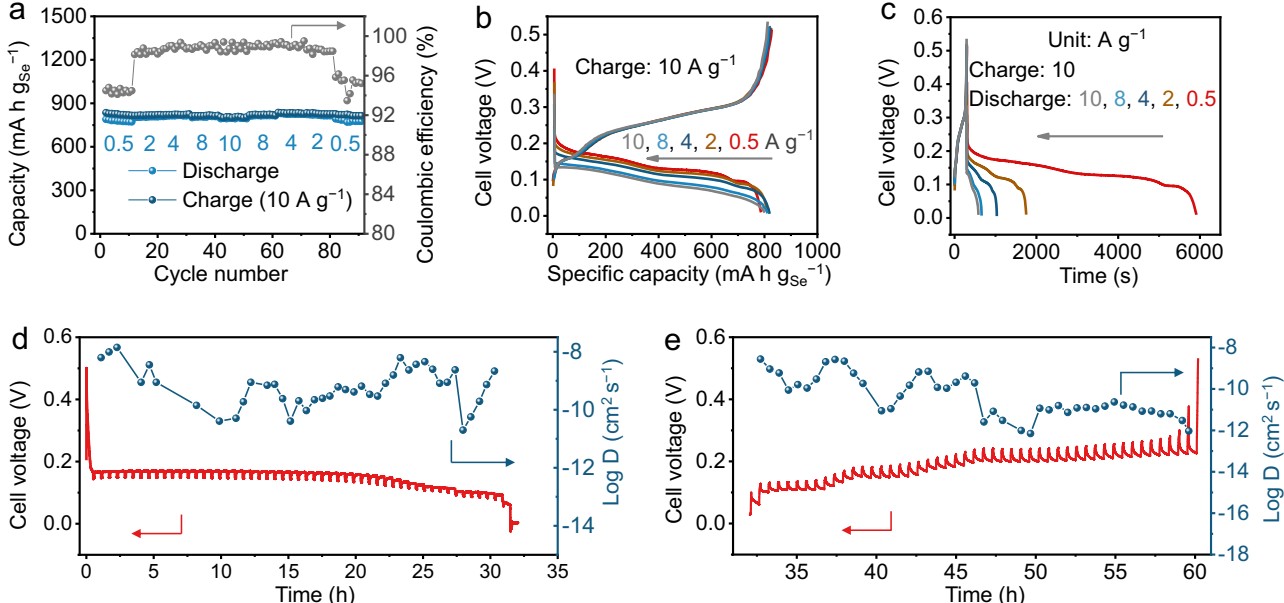

**Fig. 3 Fast-charge testing of the Cu|0.5 M CuSO₄|Se@C coin cells. a–c** The fast-charging rate performance of Cu∥Se@C-48 cell: charging at 10 A g$^{-1}$, discharging at different currents. GITT curve and the calculated Cu$^{2+}$ diffusion coefficients during (**d**) discharging and (**e**) charging processes of Cu∥Se@C-78 cell.

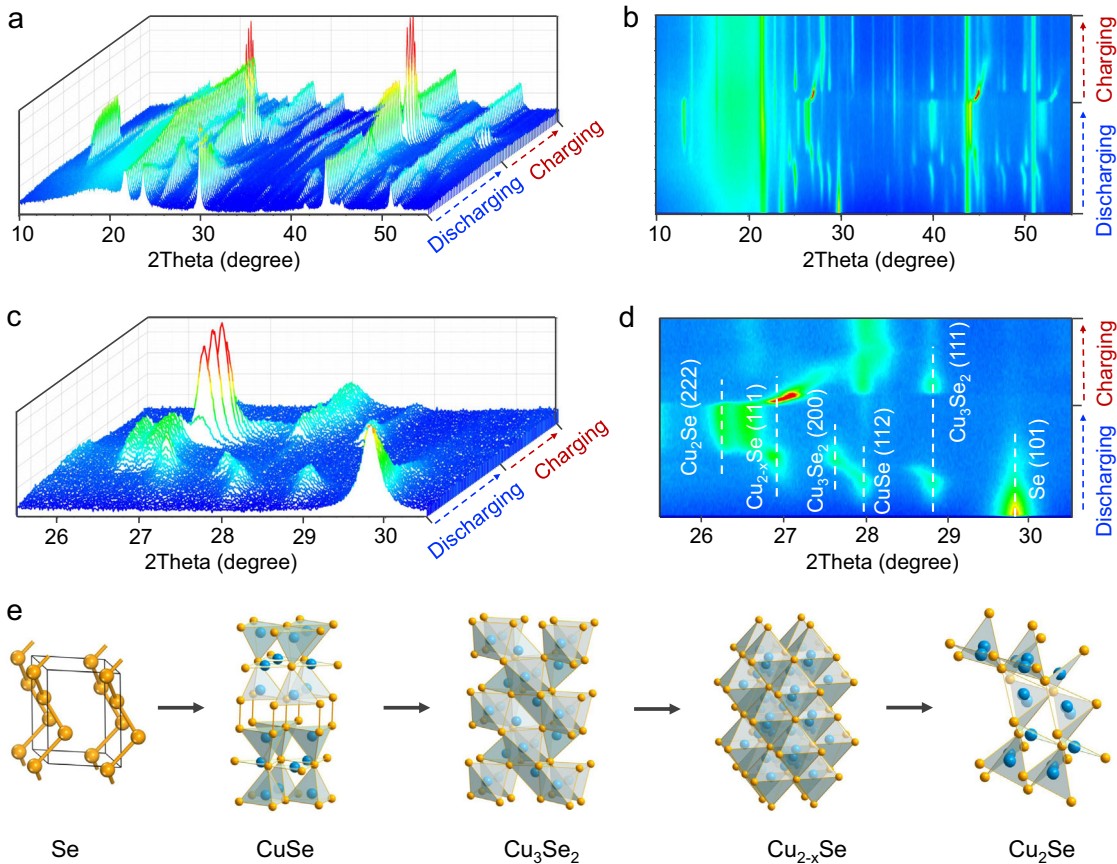

**Fig. 4 The working mechanism of the Cu|0.5 M CuSO₄|Se@C-78 coin cells. a** and **b** Full patterns of in situ XRD between 10 and 55° during a whole discharging-charging process. **c** and **d** In situ XRD patterns between 25.5 and 30.5° to show the sequential conversion of Se to CuSe, Cu₃Se₂, Cu₂₋ₓSe, and Cu₂Se during the discharging process. **e** Schematic illustration of structure transition of Se during the discharging process.

of Se. Subsequently, peaks at about 27.8°, 26.7°, and 26.3° sequentially appeared. These peaks are attributed to the (200) plane of $Cu_3Se_2$, (111) plane of $Cu_{2-x}Se$, and (222) of plane $Cu_2Se$, respectively, indicating further multi-step conversion reaction of the Se cathode. At the end of discharging, only the peak of $Cu_2Se$ remained, proving that Se was finally converted to $Cu_2Se$ in a four-electron reaction ($Se + 2\,Cu^{2+} + 4\,e^- \leftrightarrow Cu_2Se$). During the charging process, the peak of $Cu_2Se$ at $2theta$ of 26.3° vanished, and the peaks of $Cu_{2-x}Se$, $Cu_3Se_2$, and CuSe reoccurred in order. Se was also observed in the ex-situ TEM measurements of the Se@C-48 electrode at the fully charged state (Supplementary Fig. 27), suggesting the good reversibility of the cathode reaction during the discharging/charging process. The lack of Se diffraction peaks is presumably due to its low crystallinity formed at the charging state, as also reported in the literature[13,20]. The CuSe almost exists in the whole charging process, which is quite important for fast charging (*vide infra*). The discharging-charging processes of the second and third cycle were also studied by in situ XRD (Supplementary Fig. 28), demonstrating good reversibility of the four-electron Se chemistry. The ex-situ XRD was also performed, and the peaks of discharging products were matched with CuSe, $Cu_3Se_2$, $Cu_{2-x}Se$, and $Cu_2Se$ (Supplementary Fig. 29). In a word, XRD shows the Se cathode executes sequential conversions of $Se \leftrightarrow CuSe \leftrightarrow Cu_3Se_2 \leftrightarrow Cu_{2-x}Se \leftrightarrow Cu_2Se$ (Fig. 4e), delivering a four-electron transfer reaction.

The discharging products were also studied via ex-situ TEM measurements. Four discharging products (CuSe, $Cu_3Se_2$, $Cu_{2-x}Se$, and $Cu_2Se$) all show homogenous Se and Cu distribution in elemental mapping images (Fig. 5a–l). Along with

the discharging, the Cu element content in the discharging products gradually rises, suggesting the $Cu^{2+}$ charge carriers continue to react with the Se cathode. The ratio of Cu to Se elements gradually increases from 51.6: 48.4 to 40.1:59.9, 32.6: 67.4, and finally to 31.9: 68.1, which are basically in line with those of CuSe, $Cu_3Se_2$, $Cu_{2-x}Se$, and $Cu_2Se$, respectively (Fig. 5m–p and Supplementary Figs. 30–33). High-resolution TEM images of four discharging products show their inter-planar spacings of 0.322, 0.320, 0.333, and 0.206 nm that can be assigned to the (022) plane of CuSe, (200) plane of $Cu_3Se_2$, (111) plane of $Cu_{2-x}Se$, and (404) plane of $Cu_2Se$, respectively (Fig. 5q–t and Supplementary Figs. 34–38). Figures 5u–x display their corresponding fast Fourier transform (FFT) patterns. The discharging products were also investigated via ex-situ XPS electrode measurements. Supplementary Fig. 39 shows the XPS total survey spectra. On the cathodic scan, the discharging product right after the first reduction peak only showed Cu (II) 2p peaks at about 934.1 and 954.2 eV, which is in line with the valence state of Cu in CuSe (Supplementary Fig. 40a)[34]. Cu (I) 2p peaks at 931.6 and 951.8 eV begin to arise with the second reduction peak (Supplementary Fig. 40b). As the discharging process continued, the peaks of Cu (I) 2p significantly intensify and remain as the dominant feature (Supplementary Fig. 40c, d). Supplementary Fig. 41 shows the Se spectrum of discharging products at different states. All the above results support the sequential conversion of Se to CuSe, $Cu_3Se_2$, $Cu_{2-x}Se$, and finally to $Cu_2Se$ during the discharging process.

To rationalize the working mechanism of the current Se cathode, first-principle calculations were performed to elucidate the conversion process of Se and the origin of fast-charging/

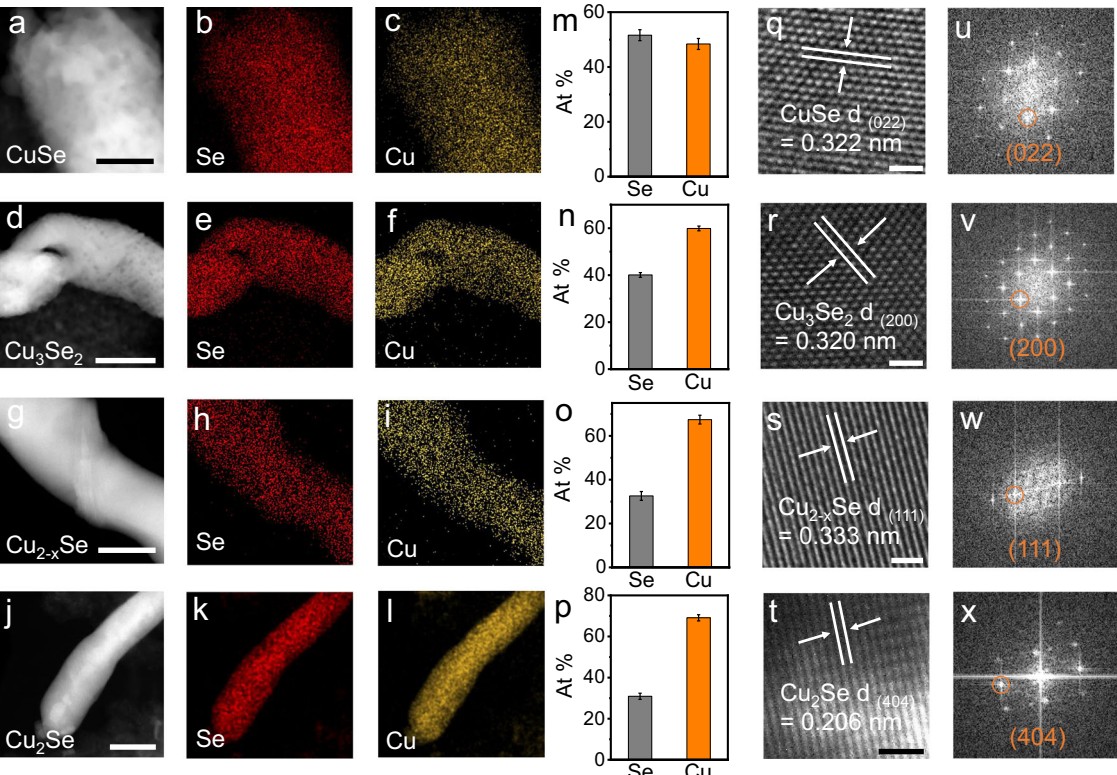

**Fig. 5 Ex situ TEM measurements of the Se@C-based electrode.** The low-resolution TEM images, corresponding element mapping images, and atomic ratio of (**a–c, m**) CuSe, (**d–f, n**) Cu$_3$Se$_2$, (**g–i, o**) Cu$_{2-x}$Se, and (**j–l, p**) Cu$_2$Se; Error bars represent the standard deviation of different experiments. The atomic ratio high-resolution TEM images and corresponding Fast Fourier transform (FFT) patterns of (**q, u**) CuSe, (**r, v**) Cu$_3$Se$_2$, (**s, w**) Cu$_{2-x}$Se, and (**t, x**) Cu$_2$Se, respectively. The scale bars are 100 nm in Fig. 5a, d, g, j, and 1 nm in 5q, r, s, t.

discharging. Figure 6a shows representative crystal structures of different discharging products of CuSe, Cu$_3$Se$_2$, Cu$_{1.75}$Se, and Cu$_2$Se. As the primitive cell of lattice changes significantly, it could be considered as a reconstructive phase transition process. However, if the positions of Se atoms are focused in these phases, a parallelepiped Se lattice could still be recognized, which serves as a lattice skeleton to include copper ions. As shown in Fig. 6a, the primitive cell of parallelepiped Se lattice is marked with red lines, and it could be found by adjusting its edge lengths and angles in all these structures, implying that the parallelepiped Se lattice deforms to provide a larger space to accommodate more copper ions. Therefore, the sequential conversion of Se to CuSe, Cu$_3$Se$_2$, Cu$_{2-x}$Se, and Cu$_2$Se in discharging can be reckoned as the consequence of a gradual filling of copper ions in the Se lattice. From theoretical perspective, the fast-charging/dischar-ging requires rapid phase transition and copper ions diffusion. For a spontaneous phase transition during discharging, the formation energy of Cu/Se alloy phases should be negative and the energy difference between Se and CuSe alloys should be higher than the thermal energy, $k_BT$, where $k_B$ and $T$ are Boltzmann constant and temperature, respectively. In fact, the formation energy is negative for all Cu/Se alloys (Supplementary Fig. 42), and the total energy difference between Se and Cu/Se alloys (~0.2 eV at zero strain, Supplementary Fig. 43, and Fig. 6b) is significantly larger than the value of thermal energy (25.9 meV at 300 K), thus thermodynamically permitting the spontaneous phase transition at the initial stage of discharging. As discharging proceeds, more and more copper ions fill into the skeleton of Se lattice, and a volumetric expansion is expected. Intuitively, the volumetric expansion could accelerate the phase transition, as it provides more room to accommodate copper ions in the lattice.

The effect of volumetric deformation on the relative total energy variation of Se, CuSe, Cu$_3$Se$_2$, Cu$_{2-x}$Se, and Cu$_2$Se is summarized in Fig. 6b. Here, the relative total energy is defined as:

$$E_i^{rel}(\varepsilon) = E_i(\varepsilon) - E_{Se}(\varepsilon), \qquad (7)$$

where $E_i^{rel}$ ($E_i$) represent relative total energy (total energy) per atom of System i ($i$ = Se, CuSe, Cu$_3$Se$_2$, Cu$_{2-x}$Se, and Cu$_2$Se), and $E_{Se}$ is total energy per atom of trigonal Se, which is taken as an energy reference. Besides, $\varepsilon$ represents equal-axis strain, and the total energy per atom continuously evolves as the equal-axis strain changes. At 10% equal-axis strain, the relative total energy $E_i^{rel}$ of Cu/Se alloys increases by ~0.1 eV, which could further promote phase transition at macroscale. Simultaneously, the rising total energy $E_i$ (around 0.5 eV) helps overcome the energy barriers between different phases; therefore the volumetric expansion due to the intercalation of copper ions during the discharging process may accelerate phase transition process as well in this aspect. A related issue is the influence of volumetric shrinking to the phase transition during the charging process. The situation here is exactly opposite to the discharging process, as copper ions are pulled out by the electric energy, all Cu/Se alloys tend to shrink with the leaving of copper ions. For all discharging products, the total energy, $E_i$, increases and the corresponding $E_i^{rel}$ decreases or even changes to a positive the sign under high volumetric shrinking (Fig. 6b), which could help speed up the conversion of Cu/Se alloys with high copper content to those with low copper content.

Additionally, the fast diffusion of copper ions is also essential to complement phase transition for rapid charging/discharging. As shown in Fig. 6c, d, Se atoms form spiral chains along the

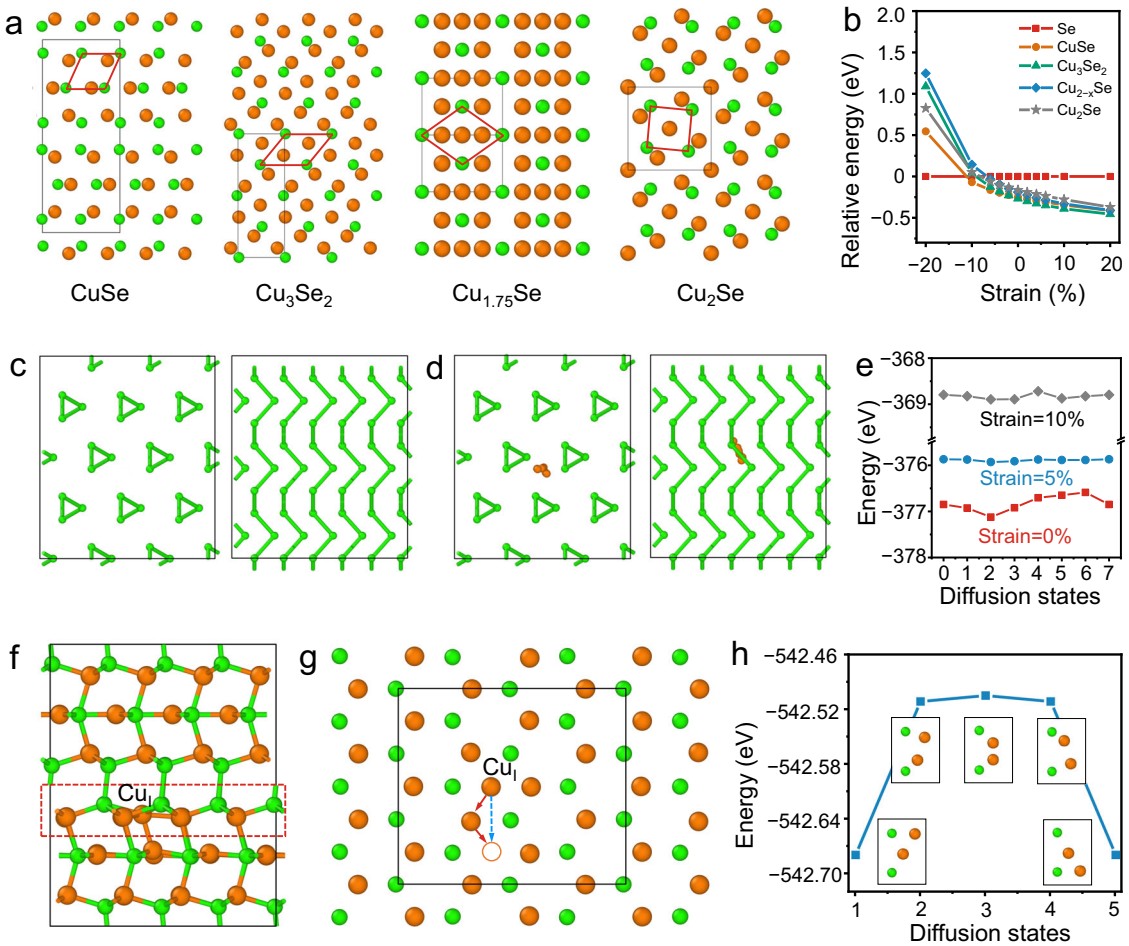

**Fig. 6 First-principle calculations of the four-electron selenium cathode chemistry. a** Atomic structures of CuSe, $Cu_3Se_2$, $Cu_{1.75}Se$, and $Cu_2Se$ and their unit cells are marked with black color, and Se parallelepiped skeleton are marked with red color; to simplify calculation, $Cu_{1.75}Se$ is used as a representative form of $Cu_{2-x}Se$. **b** The relative total energy (per atom) of Se, CuSe, $Cu_3Se_2$, $Cu_{1.75}Se$, and $Cu_2Se$ with respect to strain; The variation of particle number density over the entire process is about 69% corresponding to 19% deformation in length, therefore strain ranges from −20% to 20% in our calculation; The top and side view of crystalline Se (**d**) with or (**c**) without copper-ion embedded; Note that only one copper ion is embedded here, multiple ions appear in (**d**) are used to describe the trajectory of copper ions diffusion, and the corresponding energy variations along this trajectory under different volumetric deformation (0%, 5% and 10% equal-axis strain) are shown in (**e**). **f** The side and **g** top view of CuSe with one copper-ion embedded; In **g**, only one slice of **f** (marked with the red dash line) is shown to facilitate viewing the diffusion path of the interstitial copper ion (marked by $Cu_I$ in **f** and **g**). **h** The energy curve of copper-ion diffusion in CuSe, and the local atomic structures at different transition states are given in insets.

same direction, and it is thus assumed that the diffusion path of copper ions is along the chain axis due to the quasi-one-dimensional structure. To assess the diffusion energy barrier, the initial positions of six copper ions between two nearest neighboring Se atoms are set with equal distance from each other and structural relaxation is applied to calculate the total energy. It should be noted that during relaxation, copper ions can only move in the plane perpendicular to the chain axis. The obtained energy variation suggests a high diffusion energy barrier of ~0.54 eV at zero strain, which is difficult to trigger a fast diffusion. However, such a barrier considerably decreases under tensile strain (0.06 eV and 0.17 eV at 5% and 10% strain, respectively, Fig. 6e), suggesting the volumetric expansion contributes to fast copper ions diffusion in Se lattice. Furthermore, the diffusion of copper ions in CuSe is examined in details. As CuSe dominates the entire charging process according to XRD results (Fig. 4c, d), the fast charging should be closely related to the diffusion of copper ions in CuSe. Meanwhile, with the discharging rate primarily limited by the diffusion of copper ions through low-Cu-content domains, CuSe is the ideal model since it has the lowest copper content in all discharging products. The

model of CuSe with one copper-ion embedded is shown in Fig. 6f, g, where the diffusion process is marked by red arrows and two copper ions work collectively to render an effective diffusion process marked by the blue arrow. The energy evolution during this interstitial and vacancy pair mediated diffusion process is shown in Fig. 6h, and a relative low diffusion energy barrier of 0.174 eV is predicted, which accounts for the fast charging/discharging rates observed in the experiments. Moreover, the electrical conductivities of pure copper and all copper selenium compounds (CuSe, $Cu_3Se_2$, $Cu_{2-x}Se$, $Cu_2Se$) are calculated by first-principle method. All intermediates (CuSe, $Cu_3Se_2$, $Cu_{2-x}Se$) have decent conductivities (about one-tenth of pure copper) that could contribute to fast charging (Supplementary Fig. 44).

To demonstrate the use of the Se@C-based electrodes in other electrochemical energy storage systems, an aqueous home-made Zn||Se@C-48 full cell is assembled using an electrolyte solution comprising of 0.5 M $CuSO_4$ and 0.5 M $ZnSO_4$ (Fig. 7a). Supplementary Fig. 45 shows its photographic picture. An anion-exchange membrane is used to prevent the direct contact of the $Cu^{2+}$ ions with the Zn metal. The $SO_4^{2-}$ anions of the electrolyte serve as charge balance and are able to move through

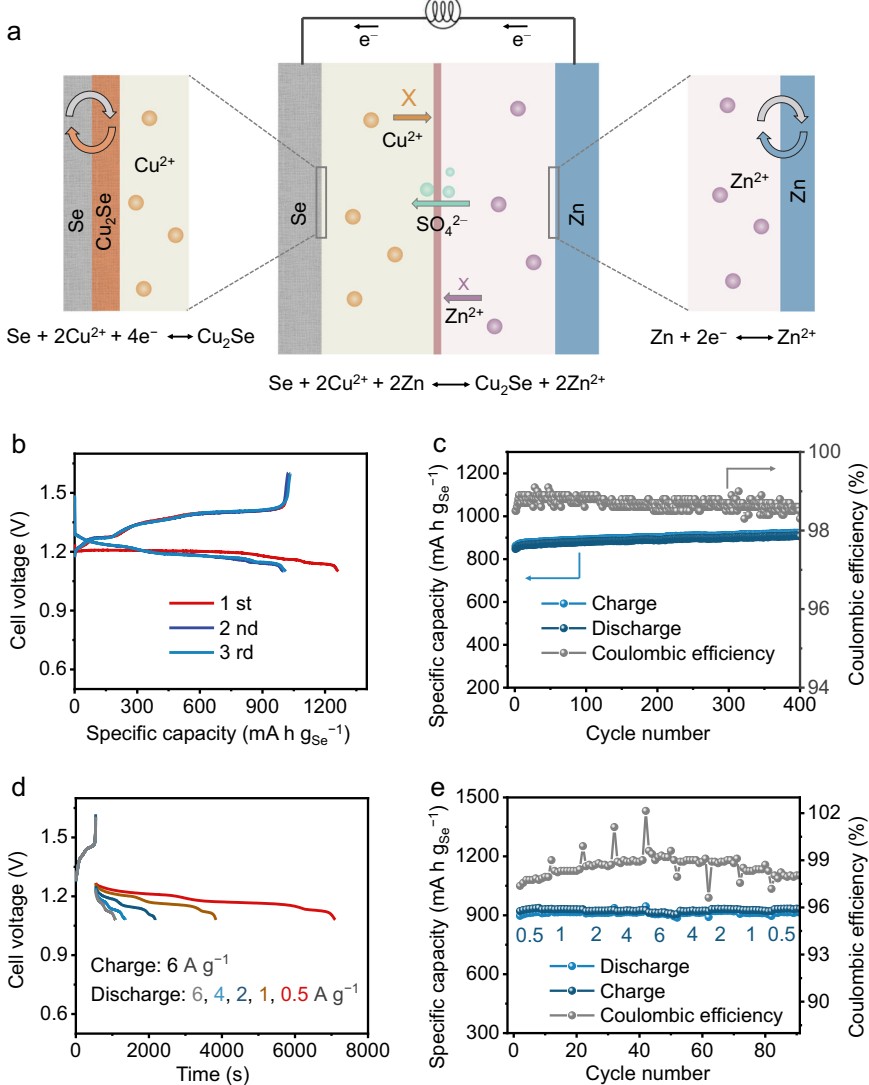

**Fig. 7 Electrochemical energy storage testing of the Zn|0.5 M ZnSO₄||0.5 CuSO₄|Se@C full cells. a** Schematic diagram of the aqueous Zn||Se@C-48 full cell. **b** GCD curves of the first three cycles at 0.5 A g⁻¹ and **c** the cycling performance at 2 A g⁻¹. **d, e** The fast-charging rate performance of the aqueous Zn||Se@C-48 full cell: charging at 6 A g⁻¹, discharging at different currents.

the membrane. During the discharging process, the Se cathode gets electrons and reacts with $Cu^{2+}$ ion to be converted to $Cu_2Se$. Concomitantly, the zinc anode loses electrons and then gets oxidized to $Zn^{2+}$. The reactions during the discharging process are as following:

$$\text{Cathode}: Se + 2Cu^{2+} + 4e^- \leftrightarrow Cu_2Se \quad (8)$$

$$\text{Anode}: 2Zn - 4e^- \leftrightarrow 2Zn^{2+} \quad (9)$$

$$\text{Full cell}: Se + 2Cu^{2+} + 2Zn \leftrightarrow Cu_2Se + 2Zn^{2+} \quad (10)$$

Benefiting from the low redox potential of Zn ($-0.76$ V vs. SHE)[35–37], the aqueous Zn||Se@C-48 full cell delivers a stable discharging voltage of about 1.2 V at 0.5 A g⁻¹. The initial discharging capacity is up to 1263 mAh g_Se⁻¹, corresponding to a specific energy of 1500 mWh g_Se⁻¹ (Fig. 7b). It delivers a reversible discharging capacity of about 1000 mAh g_Se⁻¹ in the subsequent cycles. Meanwhile, it also shows a charge/discharge hysteresis of about 0.2 V and an energy efficiency of 85.7% at 0.5 A g⁻¹. The aqueous Zn||Se@C-48 full cell could stably work for 400 cycles without significant capacity degradation at 2 A g⁻¹

(Fig. 7c). The Coulombic efficiency during cycling stability test is >98%. It could store about 900 mAh g_Se⁻¹ in 9 min (charging at 6 A g⁻¹, Fig. 7d, e), demonstrating the great potential in fast-charging applications.

## Discussion

In summary, we have reported and discussed an electrochemical energy storage system based on the $Se + 2Cu^{2+} + 4e^- \leftrightarrow Cu_2Se$ redox reaction. The Se cathode undergoes a sequential conversion of $Se \leftrightarrow CuSe \leftrightarrow Cu_3Se_2 \leftrightarrow Cu_{2-x}Se \leftrightarrow Cu_2Se$, following a four-electron transfer reaction and rendering a high theoretical specific capacity of 1350 mAh g_Se⁻¹. Besides, the Se cathode chemistry shows a redox potential of 0.5 V vs. SHE, about 1.5 V higher than those in conventional Li/Na/K–Se batteries. It possesses fast reaction kinetics and shows promising prospect in fast-charging applications. There are still some limitations of this work, which need to be improved in future research. The electrochemical performance with a high Se areal loading (>15 mg cm⁻²) requires further optimization. The anion-exchange membrane is required in Zn||Se@C-48 full cell, which leads high costs.

## Methods

**Materials**. All materials, including selenium (Se) (Aladdin, 99.9%), $CuSO_4$ (Aladdin, 99.0%), $ZnSO_4$ (Aladdin, 99.5%), zinc foil (Suzhou Wingrise energy technology Co. Ltd., thickness: 0.05 mm, 99.99%), Cu foil (Guangdong Canrd New Energy Technology Co.,Ltd., thickness: 9 μm, 99.8%), carbon cloth (Suzhou Wingrise energy technology Co. Ltd., HCP330N, thickness: 0.33 ± 0.02 mm, 160 ~ 180 g cm$^{-2}$), stainless steel (Guangdong Canrd New Energy Technology Co.,Ltd., 400 mesh), porous carbon (Nanjing Momentum Materials Technologies Co. Ltd.), and anion-exchange membrane (fumasep, FAB-PK-130, Suzhou Wingrise energy technology Co. Ltd.,), are used without further purification.

**Characterization of materials**. The morphology and microstructure analysis of the synthesized materials were examined using field-emission scanning electron microscopy (FESEM, Zeiss SUPRA$^{TM}$ 55 SAPPHIRE) and transmission electron microscopy (TEM, FEI TECNAI TF20, USA). The composition of the products was analyzed by energy dispersive spectroscopy (EDS, JEOL-6300F). The crystal structures were characterized by powder X-ray diffraction (XRD, Rigaku MiniFlex 600 diffractometer with Cu-Kα X-ray radiation, λ = 0.154056 nm). Raman spectra were performed by using a LabRAM HR Evolution (HORIBA Jobin Yvon, France) Raman microscope with a 532 nm laser. The content of Se in the prepared composite was estimated by Thermogravimetric analysis (TGA, Q50, USA) at a heating rate of 10 °C min$^{-1}$ under $N_2$ atmosphere. X-ray photoelectron spectroscopy (XPS) measurements were conducted by Thermo Scientific ESCALAB 250Xi electron spectrometer. $N_2$ sorption isotherms were measured at 77 K on Kubo X1000 sorption analyzer. The pore size distributions were calculated using the quenched solid density functional theory (QSDFT). For ex-situ XRD, TEM, and XPS measurements, the electrode samples were from Cu|0.5 M $CuSO_4$|Se@C-48 coin cells. Typically, Cu|0.5 M $CuSO_4$|Se@C-48 coin cells were performed CV tests at 0.02 mV s$^{-1}$ (negative scan from open-circuit voltage), disassembled after the first, second, third reduction peak, and full discharged state (0.01 V). The obtained cathodes were washed with water for three times and then dried under a vacuum at 60 °C for 12 h, which can then be used for various characterizations (XRD, TEM, and XPS). Sample holder with an inert atmosphere is not required during transport of the electrode samples to the equipment.

**Synthesis of Se@C composite**. The Se@C composite was obtained by a melt-diffusion method. Typically, the porous carbon and Se powder were first mixed with a certain mass ratio and then heated at 260 °C for 12 h in a tube furnace under an Ar atmosphere. The Se@C-48 composite was obtained at a Se-carbon mass ratio of 50:50. Se@C-40, Se@C-65, and Se@C-78 composites were obtained at Se-carbon mass ratios of 45:55, 70:30, and 80:20, respectively. The Se content in Se@C composites are lower than the added Se because some Se lost during the heating process.

**Electrochemical measurements**. The cathode was prepared by a slurry coating procedure and carbon cloth was used as the current collector. Typically, Se@C, Super-P carbon black (Alfa Aesar, 99%), and polyvinylidene fluoride (PVDF) binders (Alfa Aesar) were added in N-Methyl-2-pyrrolidone (NMP, Aladdin, 99.5%) with a weight ratio of 80:10:10 to form slurry (using a mortar, in air) was then uniformly deposited onto a carbon cloth. The cathode was obtained after dried under vacuum at 60 °C for 12 h. The electrochemical performance of the Se@C-based composite working electrodes was investigated in CR2032 coin cell configuration using Cu foil as counter electrode and about 200 μL of 0.5 mol L$^{-1}$ $CuSO_4$ aqueous electrolyte solution. The copper anode also acts as the reference electrode and the reference potential of the $Cu^{2+}/Cu$ is 0.33 V vs. standard hydrogen electrode (SHE). The electrochemical performance was tested on a Land cycler (Wuhan Kingnuo Electronic Co., China), all cells were first discharged. The gravimetric specific capacities were calculated based on the mass of Se. The typical areal Se loading is about 2 mg cm$^{-2}$, and the typical electrode thickness (Se@C on carbon cloth) is about 350 μm. Higher areal Se loading of about 4, 8, 12 mg cm$^{-2}$ were also fabricated and tested for higher areal capacity. The Galvanostatic intermittent titration technique (GITT) test is performed by a series of galvanostatic discharge or charge pulses of 300 s at 500 mA g$^{-1}$ followed by 1800 s rest. Electrochemical impedance spectroscopy (EIS) was tested under AC amplitude of 5 mV at the frequency from 100 kHz to 1 Hz under the open-circuit potential (constant potential). The recording number of data points was 12 (per decade). The Zn||Se@C-48 full cells were tested in home-made sealed-cell[33], where Se@C-48 as the cathode, Zn foil as the anode, an aqueous electrolyte solution comprising of 1 mL of 0.5 mol L$^{-1}$ $CuSO_4$ and 1 mL of 0.5 mol L$^{-1}$ $ZnSO_4$, and an anion-exchange membrane (fumasep, FAB-PK-130, Suzhou Wingrise energy technology Co. Ltd.,) was employed for the compartmentalization of the $Cu^{2+}$ and $Zn^{2+}$ ions. All electrochemical energy storage tests are carried out in an environmental chamber with a temperature of 25 ± 0.5 °C.

**Calculation of theoretical volume expansion of Se cathode with different charge carriers**. Taking Li–Se battery as an example. Li–Se batteries: $Se + 2 Li^+ + 2 e^- \leftrightarrow Li_2Se$.

Assuming that there is 1 mol of Se cathode at the beginning, namely 78.9 g in mass. The density of Se is 4.8 g cm$^{-3}$. So the volume of 1 mol Se is about 78.9 g/

4.8 g cm$^{-3}$ = 16.44 cm$^{-3}$. After discharging, theoretically, 1 mol of Se will convert into 1 mol of $Li_2Se$. The volume of 1 mol $Li_2Se$ is (6.9 * 2 + 78.9) g/ 2.9 g cm$^{-3}$ = 32 cm$^{-3}$. Thus, the theoretical volume expansion of Se in Li–Se batteries is 32 cm$^{-3}$/16.44 cm$^{-3}$ = 195%.

Similarly, the theoretical volume expansion of Se in Na–Se, K–Se batteries, and this work are 290%, 418%, and 183%, respectively.

**Calculation of the concentration of $Se^{2-}$ ions in the four-electron Se chemistry**. The solubility product constants ($K_{sp}$) of CuSe is $7.9 \times 10^{-49}$ [30]. For CuSe, there exists the dissolution equilibrium of $CuSe \leftrightarrow Cu^{2+} + Se^{2-}$. The concentration of $Cu^{2+}$ is 0.5 mol/L. Therefore, the concentration of $Se^{2-}$ ions is about $1.6 \times 10^{-48}$ mol/L.

**Calculation of the $Cu^{2+}$ diffusion coefficient**. It was based on the following equation[38]:

$$D = \frac{4}{\pi\tau}\left(\frac{m_B V_m}{M_B S}\right)^2\left(\frac{\Delta_{E_S}}{\Delta E_\tau}\right)^2\left(\tau \ll \frac{l^2}{D}\right) \quad (11)$$

where $D$ is $Cu^{2+}$ diffusion coefficient; $m_B$, $M_B$ and $V_m$ are the mass, molecular weight and molar volume of materials (Se); $S$ is the contact area between electrolyte and electrode; $\tau$ is the charge or discharge pulses time (300 s); $\Delta_{E_S}$ is the change of the steady-state voltage of the cell over a single titration; $\Delta E_\tau$ is the cell voltage during charging or discharging at the time of current of flux; $l$ is the thickness of the electrode.

**Calculation of specific energy of Zn||Se@C-48 full cell**. It was computed based on:

$$E = Q * V \quad (12)$$

where $E$ is specific energy (mWh g$_{Se}^{-1}$); $Q$ is thespecific capacity of cathode (mAh g$_{Se}^{-1}$); and $V$ is the voltage of Zn||Se@C-48 full cell.

**Calculation of energy efficiency of Zn||Se@C-48 full cell**.

$$E_{efficiency} = \frac{Q_{discharge} * V_{discharge}}{Q_{charge} * V_{charge}} \quad (13)$$

where $E_{efficiency}$ is energy efficiency; $Q_{discharge}$ and $Q_{charge}$ are specific capacities of discharge and charge process, respectively; and $V_{discharge}$ and $V_{charge}$ are the discharge voltage and charge voltage of Zn||Se@C-48 full cell.

**First-principle calculations details**. The projector augmented wave pseudopotential method implemented in the Vienna Ab initio Simulation Package (VASP) is used to perform structural relaxation and energy calculation[39]. The energy cutoff and electronic self-consistent step convergence are set to be 300 eV and 10$^{-5}$ eV, and the structure optimization requires energy change between two steps <10$^{-3}$ eV/ atom at least. Besides, the Perdew–Burke–Ernzerhof (PBE) exchange-correlation potential is applied for all calculations[40]. Besides, in Fig. 5b the affine volumetric deformation is adopted to simulate the lattice expansion or shrinking induced by Cu ions passing in and out, which means no further structural optimization is performed, and therefore, all sublattices are uniformly deformed.

## Data availability

All data generated in this study are provided in the Source Data file and its Supplementary Information. Source data are provided with this paper.

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

## Acknowledgements

We acknowledge financial supports from NSFC (No. 21975027, 11972349, 11790292, 22035005, 52073159, 52022051, 22075165, 22109009), NSFC-MAECI (51861135202), and the Strategic Priority Research Program of the Chinese Academy of Sciences (No. XDB22040503). The authors thank Analysis and Testing Center, Beijing Institute of Technology, for material characterizations.

## Author contributions

L.Q. and Z.Z. led the team and supervised the experiments. C.D., H.C., Y.Z., Z.Z., and L.Q. conceived the idea. C.D., L.H., X.J., and Y.H. prepared materials and performed electrochemical measurements. C.D., Y.W.,X.L., X.Z., and L.S. performed the characterizations (XRD, XPS, SEM, TEM, TGA, EIS) and analyzed the corresponding data. H.C. and M.X. contributed to in situ XRD. F.L. contributed to the theoretical calculations. All authors discussed the results and agreed on the submission of the manuscript.

## Competing interests

The authors declare no competing interests.
