## [Peer Review File · Nature Communications]

REVIEWER COMMENTS

Reviewer #1 (Remarks to the Author):

The authors report a new 'four-electron' selenium cathode chemistry in an aqueous electrolyte, which demonstrates an ultrahigh specific capacity and sufficiently high redox potential. The proposed electrochemistry reaction, if fully justified, could be very interesting. However, at the current stage, I am not fully convinced with the proposed mechanism. I would like to re-consider the manuscript after the authors further address my following concerns.

1. The authors proposed the 4-electron mechanism as $\text{Se-CuSe-Cu}_3\text{Se}_2\text{-Cu}_{2-x}\text{Se-Cu}_2\text{Se}$. In fact, it involves 2-electron $\text{Se}^0/\text{Se}^{2-}$ (Se-CuSe) and 2-electron $\text{Cu}^{2+}/\text{Cu}^+$ ($\text{CuSe-Cu}_2\text{Se}$) redox-couples. Thermodynamically, the redox potential of $\text{Se}^0/\text{Se}^{2-}$ with Cu^{2+} as charge carriers should be similar to that with other cations as charge carriers, and much lower than the $\text{Cu}^{2+}/\text{Cu}^+$ or $\text{Cu}^{2+}/\text{Cu}^0$ redox couple. In this sense, Cu stripping/plating or $\text{Cu}^{2+}/\text{Cu}^+$ should occur first, and these redox couples actually have a similar redox potential in this case. This is completely against the experimental conclusion. The authors should justify the issue with more discussion.
2. It is also necessary to conduct the CV and GCD curves of pure porous carbon and pure Se (without porous carbon), and see if similar redox peaks or charge/discharge plateau can be detected.
3. The authors assign the discharge intermediates based on only a single XRD peak, which is somehow arbitrary. Similarly, the explanation of TEM results is also based on one crystal plane. The authors should provide more convincing results to support the discharging intermediates.
4. According to the labeling of the in-situ XRD result, the final charging product is CuSe, which means the electrode can only go through the 2-electron process during charging ($\text{Cu}_2\text{Se-CuSe}$). But the calculated specific capacity is still close-to the 4-electron process. Moreover, the authors are suggested to provide the in-situ XRD result of the following charge-discharge cycles and demonstrate the reversibility of the proposed electrochemistry.
5. One minor suggestion: some statements and comparisons in the manuscript are somehow misleading. As the electrochemistry involves the $\text{Cu}^{2+}/\text{Cu}^+$ redox couple, it makes little sense to compare with conventional Se cathode in other battery systems. Also, it is not the 'four-electron selenium cathode (sounds like $\text{Se}^{2-}/\text{Se}^{2+}$ redox couple)', but the 'four-electron selenium cathode chemistry'.

Reviewer #2 (Remarks to the Author):

This is an interesting study of a novel Zn-Se battery system facilitated by Cu(II) ions. This paper is publishable as is.

Reviewer #3 (Remarks to the Author):

Manuscript Number: NCOMMS-21-39508

In this manuscript, the authors demonstrated a Cu-Se aqueous battery with high performance with fast-charging features. However, the aim is clear, but there is insufficient evidence and discussion for the aim. Therefore, it is necessary to supplement the following comments for satisfying the quality of the nature communications.

- 1. The authors have to discuss the advantage of uses of copper anode and aqueous batteries, with comparison to Li/Na/K based batteries under organic solvents.**
- 2. The authors could suggest the clear proof and fundamental approaches about the realizing the fast-charging in this system.**
- 3. It is necessary to present changes in morphology after cycling (cathode and anode). In particular, copper anode is a material that is easily oxidized and should be suggested whether there are any problems such as corrosion after repeated cycling or structural collapse.**
- 4. The authors have to discuss the reproducibility of this experiment, whether the results obtained show similar results even when repeated experiments are conducted or whether they are results from several experiments.**
- 5. It is recommended to present the results when only Se, not the Se@C composite, is used, for clear explanation of Se conversion in this system (including the battery performance).**
- 6. There is a lack of explanation for the overall extrinsic section. For example, it is necessary to mention in detail the thickness of the electrode, the amount of the electrolyte, and the information of materials. In addition, it is necessary to provide more information on the EIS results in Figure S14 like as circuit, detailed fitting, etc.**
- 7. What is the reason of using carbon fabric as current collector? Please suggest the reason and compare the using other current collector like as metal-foil.**
- 8. In Figure. 30 of supplementary file, the XPS data about Cu was presented. To clear understand, the authors have to suggest the XPS spectra about Se.**

Manuscript ID: NCOMMS-21-39508

Title: A Rapid-charging Four-electron Selenium Cathode Chemistry

Comments from reviewer 1

The authors report a new ‘four-electron’ selenium cathode chemistry in an aqueous electrolyte, which demonstrates an ultrahigh specific capacity and sufficiently high redox potential. The proposed electrochemistry reaction, if fully justified, could be very interesting. However, at the current stage, I am not fully convinced with the proposed mechanism. I would like to re-consider the manuscript after the authors further address my following concerns.

1. The authors proposed the 4-electron mechanism as $\text{Se-CuSe-Cu}_3\text{Se}_2\text{-Cu}_{2-x}\text{Se-Cu}_2\text{Se}$. In fact, it involves 2-electron $\text{Se}^0/\text{Se}^{2-}$ (Se-CuSe) and 2-electron $\text{Cu}^{2+}/\text{Cu}^+$ ($\text{CuSe-Cu}_2\text{Se}$) redox-couples. Thermodynamically, the redox potential of $\text{Se}^0/\text{Se}^{2-}$ with Cu^{2+} as charge carriers should be similar to that with other cations as charge carriers, and much lower than the $\text{Cu}^{2+}/\text{Cu}^+$ or $\text{Cu}^{2+}/\text{Cu}^0$ redox couple. In this sense, Cu stripping/plating or $\text{Cu}^{2+}/\text{Cu}^+$ should occur first, and these redox couples actually have a similar redox potential in this case. This is completely against the experimental conclusion. The authors should justify the issue with more discussion.

Response: Thanks for the comment. Thermodynamically, the standard electrode potential refers to the equilibrium potential measured where the effective concentration of redox species (eg. metal cations, counter-anions, etc.) is 1 mol L^{-1} (that is, the activity is 1). The electrode potential under a non-standard state could be derived from the Nernst equation. For instance, the actual redox potential of $\text{Se}^0/\text{Se}^{2-}$ couple, $E_{\text{Se}^0/\text{Se}^{2-}}$, varies with the concentration of free selenide concentration, $[\text{Se}^{2-}]$, and can be related to the standard redox potential of $\text{Se}^0/\text{Se}^{2-}$, $E_{\text{Se}^0/\text{Se}^{2-}}^0$, by the following equation,

$$E_{\text{Se}^0/\text{Se}^{2-}} = E_{\text{Se}^0/\text{Se}^{2-}}^0 + \frac{0.059}{2} \lg \frac{1}{[\text{Se}^{2-}]}$$

Consequently, the cations in the catholyte do play a role in determining the actual redox potential of $\text{Se}^0/\text{Se}^{2-}$ couple, $E_{\text{Se}^0/\text{Se}^{2-}}$. When $\text{Li}^+/\text{Na}^+/\text{K}^+$ serve as charge carriers, $E_{\text{Se}^0/\text{Se}^{2-}}$ is fairly close to $E_{\text{Se}^0/\text{Se}^{2-}}^0$ since the selenides of $\text{Li}^+/\text{Na}^+/\text{K}^+$ are rather soluble. However, when Cu^{2+} is selected as the charge carrier, the CuSe intermediate is highly insoluble ($K_{\text{sp}} = 7.9 \times 10^{-49}$, *Chem. Geol.* **171**, 173-194 (2001)), and $[\text{Se}^{2-}]$ is as low as about $1.6 \times 10^{-48} \text{ mol L}^{-1}$ (the concentration of Cu^{2+} is 0.5 mol L^{-1}). Using this value, the redox potential of this novel Se cathode, $E_{\text{Se}^0/\text{Se}^{2-}}$, is $\sim 1.42 \text{ V}$ higher than $E_{\text{Se}^0/\text{Se}^{2-}}^0$. A similar enhancement in the redox potential was also observed while Zn^{2+} as the charge carriers because ZnSe is also insoluble (*Energy Environ. Sci.* **14**, 2441-2450 (2021)).

Therefore, the redox potential of novel Se cathode chemistry (about 0.5 V vs. SHE) is higher than that of Cu^{2+}/Cu (0.340 vs. SHE) or $\text{Cu}^{2+}/\text{Cu}^+$ (0.159 vs. SHE) couples. We have now added the corresponding discussion in the introduction part (Page 3) of the revised manuscript.

2. It is also necessary to conduct the CV and GCD curves of pure porous carbon and pure Se (without porous carbon), and see if similar redox peaks or charge/discharge plateau can be detected.

Response: As suggested, we have now added the CV and GCD curves of pure Se and pure carbon as Supplementary Figures 11 and 13, respectively. We also added “The bare Se delivers similar CV and GCD curves as the Se@C cathode, but showing lower specific capacity and inferior rate performance due to large size of bare Se particles (Supplementary Figures 11 and 12). The pure carbon shows negligible specific capacity (about 1.5 mAh g^{-1}) in this system (Supplementary Figure 13).” in Page 5 in the revised manuscript.

Supplementary Figure 11. The electrochemical performance of pure Se. The **a** CV curves at 0.02 mV s^{-1} , **b** GCD curves at 500 mA g^{-1} , **c** rate performance, and **d** cycling performance of pure Se as the cathode. The first discharge specific capacity of pure Se cathode is $910 \text{ mAh g}_{\text{Se}}^{-1}$.

Supplementary Figure 12. SEM image of pure Se.

Supplementary Figure 13. The typical GCD and CV curves of pure porous carbon as the cathode. The pure porous carbon shows a negligible specific capacity of about 1.5 mAh g^{-1} .

3. The authors assign the discharge intermediates based on only a single XRD peak, which is somehow arbitrary. Similarly, the explanation of TEM results is also based on one crystal plane. The authors should provide more convincing results to support the discharging intermediates.

Response: The discharge intermediates were also scrutinized by the *ex-situ* XRD (Supplementary Figure 28). All peaks of discharging products at B, C, D, and E in Supplementary Figure 28a are well matched with CuSe , Cu_3Se_2 , Cu_{2-x}Se , and Cu_2Se , respectively.

As for the TEM, in addition to crystal plane in Figure 4, we have now provided other crystal planes of discharge intermediates, please see Supplementary Figure 37.

In general, *In-situ* XRD, *ex-situ* XRD, TEM, EDS spectrum, and XPS results all indicate the sequential conversions of $\text{Se} \leftrightarrow \text{CuSe} \leftrightarrow \text{Cu}_3\text{Se}_2 \leftrightarrow \text{Cu}_{2-x}\text{Se} \leftrightarrow \text{Cu}_2\text{Se}$, proving the presence of a four-electron transfer chemistry in the current cathode.

Supplementary Figure 28. *Ex situ* XRD patterns of discharging products. **a** The typical CV curve of Se@C-48 at a scan rate of 0.02 mV s⁻¹. XRD patterns of sample **b** before discharging, at **c** the first reduction peak, **d** second reduction peak, **e** third reduction peak, and **f** 0.01 V vs. Cu²⁺/Cu.

Supplementary Figure 37. Other high-resolution TEM images of four discharging products show their inter-planar spacings of 0.319, 0.311, 0.203, and 0.338 nm that can be assigned to the **a** (112) plane of CuSe, **b** (111) plane of Cu_3Se_2 , **c** (220) plane of Cu_{2-x}Se , and **d** (222) plane of Cu_2Se , respectively.

4. According to the labeling of the in-situ XRD result, the final charging product is CuSe, which means the electrode can only go through the 2-electron process during charging (Cu_2Se -CuSe). But the calculated specific capacity is still close-to the 4-electron process. Moreover, the authors are suggested to provide the in-situ XRD result of the following charge-discharge cycles and demonstrate the reversibility of the proposed electrochemistry.

Response: Thanks for the comment. It is worth noting that the *in-situ* XRD was

performed at a relatively large current density (200 mA g^{-1}), and it is probable that some CuSe is not completely converted to Se. The CuSe is quite important for fast-charging.

The TEM images of the final charging product suggest some Se is formed. The lack of obvious diffraction peaks is presumably due to its low crystallinity after cycling, and this phenomenon is also observed in other Se-based batteries (*Energy Environ. Sci.* **14**, 2441-2450 (2021); *ACS Sustain. Chem. Eng.* **6**, 7064-7077 (2018)). Therefore, it is reasonable that specific capacity is close to the 4-electron process. We have now added “Se was also observed in the TEM image of the final charging product (Supplementary Figure 26), suggesting the good reversibility of the cathode reaction during the discharging/charging process. The lack of obvious Se diffraction peaks is presumably due to its low crystallinity formed at the charging state, which is also observed in other Se-based batteries.”^{13, 20} in Page 8 in the revised manuscript, and provided the TEM image of the final charging product as Supplementary Figure 26.

As suggested, we have now also provided the *in-situ* XRD result of the following charge-discharge cycles as Supplementary Figure 27. The periodic appearance and disappearance of CuSe, Cu_3Se_2 , Cu_{2-x}Se , and Cu_2Se during the discharging and charging processes suggesting the good reversibility of the four-electron Se chemistry. We have now added “The discharging-charging processes of the second and third cycle were also studied by *in-situ* XRD (Supplementary Figure 27), demonstrating good reversibility of the four-electron Se chemistry.” in Page 8 in the revised manuscript.

Supplementary Figure 26. TEM image of the final charging product in the *in-situ* XRD measurements. Se is observed in the final charging product, suggesting the reversibility of the four-electron Se chemistry.

Supplementary Figure 27. *In-situ* XRD of the second and third cycle discharging-charging process; **a-b** show the full patterns between 10 and 55°. **c-d** show XRD patterns between 25.5 and 29° to clearly present the sequential conversion of Se to CuSe, Cu₃Se₂, Cu_{2-x}Se, and Cu₂Se during the discharging process.

5. One minor suggestion: some statements and comparisons in the manuscript are somehow misleading. As the electrochemistry involves the Cu²⁺/Cu⁺ redox couple, it makes little sense to compare with conventional Se cathode in other battery systems. Also, it is not the ‘four-electron selenium cathode (sounds like Se²⁻/Se²⁺ redox couple)’, but the ‘four-electron selenium cathode chemistry’.

Response: As suggested, we have now changed the statements of “four-electron selenium cathode” into “four-electron selenium cathode chemistry” and changed the title into “A Rapid-charging Four-electron Selenium Cathode Chemistry”.

Comments from reviewer 2

This is an interesting study of a novel Zn-Se battery system facilitated by Cu(II) ions. This paper is publishable as is.

Response: We would like to thank the reviewer for reviewing the manuscript.

Comments from reviewer 3

In this manuscript, the authors demonstrated a Cu-Se aqueous battery with high performance with fast-charging features. However, the aim is clear, but there is insufficient evidence and discussion for the aim. Therefore, it is necessary to supplement the following comments for satisfying the quality of the nature communications.

1. The authors have to discuss the advantage of uses of copper anode and aqueous batteries, with comparison to Li/Na/K based batteries under organic solvents.

Response: The copper anode is used to study the novel Se chemistry in the coin cell. The voltage output is relatively low when copper anode is used. In order to demonstrate the application of the novel Se chemistry, an aqueous Zn-Se hybrid battery is constructed (Figure 6) and it rivals best performance of the state-of-the-art aqueous Zn-based batteries.

In the introduction part (Page 3) of the revised manuscript, we have now discussed the advantages of this aqueous Se chemistry, including high redox potential, impressive specific capacity, and fast-charging performance compared to Li/Na/K-Se batteries under organic electrolytes.

2. The authors could suggest the clear proof and fundamental approaches about the realizing the fast-charging in this system.

Response: Thanks for the comment. First, the electrochemical performance shows this novel Se chemistry shows much better rate performance than other Se-based batteries (Figure 2h). Second, the galvanostatic intermittent titration experiments (GITT) show the typical gaps between each polarization potential and each quasi-equilibrium potential are as small as 18.9 mV for the charging processes, (Supplementary Figure 21), suggesting the fast reaction kinetics. Third, the Cu^{2+} diffusion coefficients are in the ranges of 10^{-12} to 10^{-8} $\text{cm}^2 \text{s}^{-1}$ for the charging processes (Figure 2e and 2f), which is faster than the Li^+ diffusion coefficient in

high-rate black phosphorus composites ($7.5 \times 10^{-13} \text{ cm}^2 \text{ s}^{-1}$, Science 370, 192–197 (2020)). Fourth, first principle calculations show the Cu^{2+} diffusion energy barrier is as low as 0.174 eV, which accounts for the fast charging/discharging rates observed in the experiments (Figure 5h). Last, all intermediates (CuSe , Cu_3Se_2 , Cu_{2-x}Se) have decent conductivities (about one tenth of pure copper) that could contribute to fast-charging.

We have now added “Moreover, the electrical conductivities of pure copper and all copper selenium compounds (CuSe , Cu_3Se_2 , Cu_{2-x}Se , Cu_2Se) are calculated by first principle method. All intermediates (CuSe , Cu_3Se_2 , Cu_{2-x}Se) have decent conductivities (about one tenth of pure copper) that could contribute to fast-charging (Supplementary Figure 43).” In Page 14 in the revised manuscript.

Supplementary Figure 43. Density of states (DOS) per unit cell and conductivity for **a-b** Cu, **c-d** CuSe, **e-f** Cu₃Se₂, **g-h** Cu_{1.75}Se, and **i-j** Cu₂Se. Vienna Ab-initio Simulation Package (VASP) is used to obtain band structures followed by conductivity calculation implemented in BoltzTrap2^{S1} based on Boltzmann transport equation. Relaxation time takes 10^{-14} s in all conductivity calculations.

- It is necessary to present changes in morphology after cycling (cathode and anode).
In particular, copper anode is a material that is easily oxidized and should be

suggested whether there are any problems such as corrosion after repeated cycling or structural collapse.

Response: As suggested, we now have provided the morphology of cathode and anode after cycling as Supplementary Figure 9 and 10 in the Supporting Information. We also added “After long cycling, the structure of cathode is well maintained and some Cu_2O particles appear on Cu anode due to dissolved oxygen in electrolyte (Supplementary Figure 9 and 10).³²” in Page 5 in the revised manuscript.

Supplementary Figure 9. SEM images of Se@C-48 electrode **a-c** before and **d-e** after cycled.

Supplementary Figure 10. **a** The XRD pattern and **b-c** SEM images of Cu anode before cycled. **d** The XRD pattern and **e-f** SEM images of Cu anode after cycled 300 cycles at 5 A g^{-1} .

The fresh Cu anode is well matched with Cu (JCPDS No. 04-0836) and shows a rough plane structure. After cycled for 300 cycles at 5 A g^{-1} , it changed into Cu/Cu₂O composite. The formation of Cu₂O is mainly attributed to the dissolved oxygen in

electrolyte. The SEM images show that some irregular particles with tens of micrometers in size were formed, which should be attributed to the Cu/Cu₂O composite.

4. The authors have to discuss the reproducibility of this experiment, whether the results obtained show similar results even when repeated experiments are conducted or whether they are results from several experiments.

Response: The repeatability of the results is good. As suggested, we have now added error bars in Figure 1b. Error bars represent the standard deviation of five independent batteries.

Figure 1. Electrochemical performance of the novel Se chemistry. **a** GCD curves of the first three cycles of Se@C-48 (the number represents the Se content) at 0.5 A g⁻¹; **b** The redox potential and specific capacity comparison with other Se-based cathodes; **Error bars represent the standard deviation of five independent batteries.** **c** The ultra-stable discharging plateau and ultra-low polarization between discharging and charging curves of Se@C-48 at 1 A g⁻¹. The comparisons of **d** discharging slope and **e** polarization with other Se-based cathodes. The slopes are obtained through fitting the data and the smaller slope means the more stable discharging plateau. **f** Rate performance and **g** corresponding GCD curves. **h** The cycling performance of Se@C-48 cathode at 5 A g⁻¹.

5. It is recommended to present the results when only Se, not the Se@C composite, is used, for clear explanation of Se conversion in this system (including the battery performance).

Response: As suggested, we have now studied the electrochemical performance of pure Se and added it as Supplementary Figure 11.

Pure Se cathode also shows similar CV and GCD curves, suggesting it delivers the same electrochemical performance. However, pure Se features a large size of about several micrometers, which results in long ion transport distance and slow mass transfer, thus leading to a reduced rate performance. Besides, almost all previous reported work about Se-based batteries adopted various carbon as the host and studied the electrochemical performance of Se@C composites.

Supplementary Figure 11. The electrochemical performance of pure Se. The **a** CV curves at 0.02 mV s^{-1} , **b** GCD curves at 500 mA g^{-1} , **c** rate performance, and **d** cycling performance of pure Se as the cathode. The first discharge specific capacity of pure Se cathode is $910 \text{ mAh g}_{\text{Se}}^{-1}$.

Supplementary Figure 12. SEM image of pure Se.

6. There is a lack of explanation for the overall extrinsic section. For example, it is necessary to mention in detail the thickness of the electrode, the amount of the electrolyte, and the information of materials. In addition, it is necessary to provide more information on the EIS results in Figure S14 like as circuit, detailed fitting, etc.

Response: Thanks for the suggestion. The typical electrode thickness (Se@C on carbon cloth) is about 350 μm. The typical amount of the electrolyte for the CR2032 coin cell is about 200 μL. Electrochemical impedance spectroscopy (EIS) was tested under AC amplitude of 5 mV at the frequency from 100kHz to 1 Hz under the open-circuit potential (constant potential). The recording number of data points was 12 (per decade).

We have now added the relevant details in Electrochemical measurements part in Pages 16-17 of the revised manuscript. The equivalent circuit and fitted curves are also provided in Supplementary Figure 20.

Supplementary Figure 20. The EIS comparison of Se@C-40, Se@C-48, Se@C-65, and Se@C-78. Inset shows the corresponding equivalent circuit. The points and lines represent measured and fitted curves, respectively.

7. What is the reason of using carbon fabric as current collector? Please suggest the reason and compare the using other current collector like as metal-foil.

Response: Thanks for the comments. Many published works about aqueous batteries (*Nat. Commun.* 2019, 10, 3227; *Adv. Energy Mater.* 2019, 9, 1901838; *Adv. Energy Mater.* 2020, 10, 1904163; *Adv. Energy Mater.* 2020, 10, 1903589; *Adv. Mater.* 2021, 33, 2105480) adopted carbon fabric as the current collectors mainly because of its excellent electric conductivity, corrosion resistance and compatibility with aqueous electrolytes. Another commonly used current collector in aqueous batteries is stainless steel mesh (*Nat. Commun.* 2021,12, 6878; *Nat. Commun.* 2021, 12, 2857 ; *Nat. Commun.* 2019, 10, 4948). We also studied the electrochemical performance of Se@C cathode with stainless steel mesh as the current collector, and it shows similar CV and GCD curves.

We have now added “The cells with carbon cloth and stainless steel as the current collectors show similar cyclic voltammetry (CV) and galvanostatic charge-discharge (GCD) curves (Supplementary Figure 6), and carbon cloth is adopted in the subsequent tests.” in Page 4 in revised manuscript.

Supplementary Figure 6. The batteries with stainless steel and carbon cloth as the current collectors show similar CV and GCD curves.

8. In Figure. 30 of supplementary file, the XPS data about Cu was presented. To clear understand, the authors have to suggest the XPS spectra about Se.

Response: As suggested, we have now provided the XPS spectra of Se as Supplementary Figure 40. All Se spectrum show three main peaks, corresponding to $\text{Se } 3d_{5/2}$, $3d_{3/2}$, and SeO_x , respectively, and are consistent with previous works (*Appl. Surf. Sci.* **466**, 401-410 (2019); *Chem. Eng. J.* **384**, 123235 (2020); *J. Mater. Chem. A* **9**, 3648-3656 (2021)). The Cu and Se spectrum both support the sequential conversion of Se to CuSe , Cu_3Se_2 , Cu_{2-x}Se , and finally to Cu_2Se during the discharging process.

Supplementary Figure 40. High-resolution XPS spectra study of discharging products

at different states. Se peaks of **a** CuSe, **b** Cu₃Se₂, **c** Cu_{2-x}Se, and **d** Cu₂Se. All Se spectrum show three main peaks, corresponding to Se 3d_{5/2}, 3d_{3/2}, and SeO_x, respectively.

REVIEWERS' COMMENTS

Reviewer #1 (Remarks to the Author):

Overall, the authors have addressed all my concerns very well. I recommend accepting the manuscript as it is.

Reviewer #3 (Remarks to the Author):

The reviewer's questions were answered well enough. Therefore, I recommend this article to nature communications.